# Effect of host fruit, temperature and *Wolbachia* infection on survival and development of *Ceratitis capitata* immature stages

**Niki K. Dionysopoulou, Stella A. Papanastasiou, Georgios A. Kyritsis, Nikos T. Papadopoulos**  *

Laboratory of Entomology and Agricultural Zoology, Department of Agriculture Crop Production and Rural Environment, University of Thessaly, Volos, Greece

* nikopap@uth.gr

## Abstract

The Mediterranean fruit fly, *Ceratitis capitata* (Diptera: Tephritidae), holds an impressive record of successful invasions promoted by the growth and development of international fruit trade. Hence, survival of immatures within infested fruit that are subjected to various conditions during transportation seems to be a crucial feature that promotes invasion success. *Wolbachia pipientis* is a common endosymbiont of insects and other arthropods generating several biological effects on its hosts. Existing information report the influence of *Wolbachia* on the fitness traits of insect host species, including the Mediterranean fruit fly. However, little is known regarding effects of *Wolbachia* infection on immature development in different host fruits and temperatures. This study was conducted to determine the development and survival of immature stages of four different Mediterranean fruit fly populations, either infected or uninfected with *Wolbachia*, in two hosts (apples, bitter oranges) under three constant temperatures (15, 25 and 30˚C), constant relative humidity (45–55 ± 5%), and a photoperiod of 14L:10D. Our findings demonstrate both differential response of two fruit fly lines to *Wolbachia* infection and differential effects of the two *Wolbachia* strains on the same Mediterranean fruit fly line. Larva-to-pupa and larva-to-adult survival followed similar patterns and varied a lot among the four medfly populations, the two host fruits and the different temperatures. Pupation rates and larval developmental time were higher for larvae implanted in apples compared to bitter oranges. The survival rates of wildish medflies were higher than those of the laboratory adapted ones, particularly in bitter oranges. The *Wolbachia* infected medflies, expressed lower survival rates and higher developmental times, especially the *w*Cer4 infected line. High temperatures constrained immature development and were lethal for the *Wolbachia* infected *w*Cer4 medfly line. Lower temperatures inferred longer developmental times to immature stages of all medfly populations tested, in both host fruits. Implications on the ecology and survival of the fly in nature are discussed.

**Data Availability Statement:** All relevant data are provided as a Supporting Information file.

**Funding:** This research is financed by the General Secretariat for Research and Technology (GSRT)

and Hellenic Foundation for Research and Innovation (HFRI).

**Competing interests:** The authors have declared that no competing interests exist.

## Introduction

The distribution and abundance of true fruit flies (Diptera: Tephritidae) depend on several abiotic (e.g., temperature, relative humidity, rainfall) and biotic factors (e.g., host plants, natural enemies) [1]. The Mediterranean fruit fly (medfly), *Ceratitis capitata* (Wiedemann) (Diptera: Tephritidae) is considered one of the most devastating pests of fruit and vegetables worldwide [2–4]. It is a highly polyphagous insect pest infesting more than 300 plant species [5,6]. Moreover, medfly is a remarkably invasive pest, currently holding an almost world-wide geographical distribution [7], and expressing an enormous ability to cope with diverse biotic and abiotic factors. In the long list of hosts, some are highly favorable (citrus), while others are marginal (apples) for immature development and adult performance. In temperate areas, high infestations of pome and stone fruit, as well as of lower economic value hosts such as figs and persimmons during summer, lead to the development of high medfly populations in autumn. These populations threaten later maturing hosts such as apples. However, development in apples, which are considered not favorable for medfly, may have a crucial role for the fly's successful overwintering as larva within the fruit, and hence it's persistence in temperate areas [8–10].

*Wolbachia pipientis* is a gram-negative, obligate intracellular bacterium belonging to the family of Rickettsiaceae (α-proteobacteria) that infects numerous species of arthropods and nematodes [11]. It has been estimated that up to 66% of insect species are infected with *Wolbachia* [12]. It is maternally inherited and capable of manipulating host reproduction favoring its own dispersion as a reproductive parasite [13]. Cytoplasmic incompatibility (CI) is the most common *Wolbachia* induced phenotype in insects and mites, which leads to embryonic mortality of fertilized eggs when an infected male mates with i) an uninfected female (unidirectional CI), or ii) a female infected with a different *Wolbachia* strain (bidirectional CI) [14]. Furthermore, *Wolbachia* infection may also affect the olfactory response, life span, and immunity of its hosts [15,16].

*Wolbachia* transmission is directly depended on the survival and reproduction of its host; hence, it is rather expected that the *Wolbachia*-insect symbiosis would promote host's fitness. Indeed, several studies confirm this assumption demonstrating positive effects of *Wolbachia* infection on the fitness traits of insect hosts, such as increased longevity and improved reproduction rates [17–21]. However, other studies report significant fitness costs on infected populations [22–25]. Differential fitness responses among *Wolbachia* infected insect hosts could be attributed to the rate that a mutualistic relationship between the insect and the bacterium is evolving. Such an example has been recorded in *Drosophila simulans* wild populations which can be found either infected or not with *Wolbachia*. Although, uninfected females exhibited higher egg-production rates than infected ones, this trend has shifted in less than 20 years of bacterium-insect co-evolution [26], indicating the dynamic nature of the symbiotic relationships, and possibly the neglected role of the abiotic factors that when interact with a given biotic environment could differentiate specific fitness characters.

Environmental conditions, and especially temperature, affect the survival and fitness of insects [27–31]. Tephritids can survive and develop within a range of temperatures with the extreme ones functioning as tolerance thresholds [28,32]. The demographic response of tephritids to different temperatures has been extensively studied under both constant and fluctuating temperatures [33–40] with considerable attention given in the development of immature stages [41–43]. Effects of host fruits on immature survival and developmental duration have been tested mainly under optimal environmental conditions (e.g. constant temperature and humidity) [1,43–45]. Additional studies, focusing on quarantine treatments have determined mortality rates of immatures after being exposed within artificially infested fruit at low storage temperatures for extended periods of time [46–48].

In a list of interesting studies, Diamantidis et al. [49–51] demonstrated that medfly populations obtained from different geographic areas express different life history traits under the same constant laboratory conditions. Other studies have confirmed that laboratory adapted and wild or wildish populations (reared in artificial diet for up to 10–12 generations under laboratory conditions) of the Mediterranean fruit fly may express different demographic profiles [45,49]. Over long periods of laboratory rearing that induce rather directed selection pressures, *C. capitata* populations go through bottlenecks that reduce the genetic variability, the diversity and possible abundance of their microbiome. Although this process results in an increased fitness under the artificial rearing conditions, laboratory adapted flies perform poorly under different rearing conditions or in the wild [52,53]. Laboratory adapted populations, therefore, may be unable to develop in wild hosts and in variable environmental conditions. The interaction of factors such as temperature, host fruit and different medfly populations may contribute towards understanding the plastic and adaptive aspects of development and performance of medflies.

Over the last decade a growing body of studies has explored the *Wolbachia* infection status of many important species of Tephritidae [54]. Wild populations from a list of species such as *Bactocera dorsalis*, *Anastrepha suspensa*, *Rhagoletis cerasi*, *Zeugodacus cucurbitae* have been found to be infected with *Wolbachia* [55–58]. Nonetheless, and despite intense efforts all *C. capitata* populations tested were found to be *Wolbachia* free [59]. Hence, trans-infection of laboratory adapted populations offered an interesting opportunity to establish a CI tool to control this pest. The earlier studies of Boller et al. [60,61] that demonstrated CI in wild populations of the European cherry fruit fly, *R. cerasi* (L) have been considered in these new efforts. In fact, the CI among *R. cerasi* populations that Boller and colleagues detected in 1970's has been attributed to different *Wolbachia* infections by Riegler et al. [55]. The *Wolbachia* strains *w*Cer2 and *w*Cer4 originated from *R. cerasi* (donor) have been successfully transferred to laboratory adapted *C. capitata* lines [62]. Crosses between *Wolbachia* infected males and non-infected females of *C. capitata*, under controlled laboratory conditions, resulted in strong CI (100% embryonic mortality) [62]. In addition, *w*Cer2 and *w*Cer4 expressed bidirectional CI in *C. capitata* [62,63]. Establishment of a successful CI project against *C. capitata* based on *Wolbachia* infected laboratory adapted populations requires efficient rearing and sexing procedures as well as the production of high-quality males that can outcompete wild males to mate with wild females.

The first effort to address effects of *Wolbachia* infection on demographic traits of medfly populations has been conducted by Sarakatsanou and co-workers [64]. The respective study revealed the effects of a single *Wolbachia* strain (*w*Cer2) on fitness components of two *C. capitata* genotypes (i.e., Benakeio and Vienna 8 GSS laboratory lines), as well as the effects of two different *Wolbachia* strains (*w*Cer2 and *w*Cer4) on a single medfly genotype (Benakeio). The following general patterns emerged: a) *Wolbachia* causes high egg-to-larva mortality, b) *Wolbachia* causes high egg-to-adult mortality (exception: Vienna 8 GSS + *w*Cer2) and c) *Wolbachia* shortens egg-to-adult developmental time, although it prolongs embryonic development (exception: Benakeio + *w*Cer2). Recent studies by Kyritsis [65,66] on the same system (*Wolbachia*–medfly) reported no effects of *Wolbachia* infection on adult lifespan and a reduced fecundity in the case of *w*Cer4 infection only. Even though *w*Cer2 and *w*Cer4 in general tended to have consistent effects on medfly, the magnitude of their effects differed. Collectively, the results from these studies indicate that the effect of *Wolbachia* infection on life history traits depends both on the *C. capitata* genetic background and on the *Wolbachia* strain.

The aim of the current study was to explore the response of immature stages of different medfly populations (laboratory adapted and wildish ones) to different host fruits (apples and bitter oranges) and under different temperature regimes. Including *Wolbachia* infected lines

in our experimentation we have been interested to address the effects of *Wolbachia* on the immature development of medfly on wild fruit host. Moreover we explored whether the interaction between temperature and wild host fruits determines the performance of immature medflies.

## Materials and methods

Experiments were conducted in the laboratory of Entomology and Agricultural Zoology at the University of Thessaly from September 2017 to April 2018. The conditions in the experimental rooms were set at 25±1˚C, 45–55% relative humidity and 14:10 L:D photoperiod (photo phase started at 07:00 h). Light was provided by fluorescent tubes adjusted on the top the shelves where rearing cages were kept and experiments were performed. Light intensity on the shelves ranged from 1500 to 2000 Lux.

We used four different Mediterranean fruit fly populations: (a) 'Benakeio', a *Wolbachia* free (uninfected) laboratory population, (b) 88.6, a trans-infected Benakeio line carrying the *w*Cer2 *Wolbachia* strain [62], (c) S10.3, a trans-infected Benakeio line carrying the *w*Cer4 *Wolbachia* strain [62] and (d) one wildish population originated from field infested apples collected in Agia, Larissa that has been reared in laboratory conditions for 9–12 generations.

Apple (Golden Delicious) and bitter orange (local cultivar) fruits, collected from organic orchards located in central Greece (Thessaly district), were used in our experiments. No endangered or protected species were involved and no specific permission was required for the location nor the activities carried out. The permissions of the organic orchards' owners to enter the properties and collect fruits were acquired before starting the experiments. Both host fruits, stored at 6 ± 2˚C for 2 to 14 days, were thoroughly washed with tap water before being used in our tests.

The rearing procedure of the four *C. capitata* populations was performed by caging approximately 100 adults in wooden, (30 x 30 x 30 cm), wire-screened cages with *ad libidum* access to water and standard adult diet (mixture of yeast hydrolysate, sugar and water at 1:4:5 ratio— YS). Females deposited eggs on the inner surface of artificial oviposition substrates [5cm diameter hollow, plastic, red-colored hemispheres (domes), pin-punctured with 40–50 evenly distributed holes (1mm diameter)]. Domes were fitted in the cover of a 5.5-cm diameter plastic petri dish. Water was added in the base of the petri dish to maintain high humidity levels within the dome. Also, 0.5 ml of orange juice was added in a small plastic cup placed in the base of the petri dish to stimulate oviposition [67,68]. Domes were placed in rearing cages for 24h in order to collect enough eggs to continue the rearing or to use in experiments.

To investigate the effects of *Wolbachia* infection, fruit-host and temperature on the immature (larvae, pupae) survival and developmental duration, freshly laid eggs taken from the oviposition domes were spread using a soft paintbrush on a black filter paper, impregnated with water and fitted in a Petri dish (4.5 cm in diameter). A small piece of apple (2x2 cm) was placed on the center of the filter paper to attract newly hatched larvae and prevent them from scattering within the Petri dish. Filter papers with eggs were kept moist for 48h and subsequently first instar larvae were hatched. Freshly hatched larvae were implanted in two artificial holes 1.5–2.0 mm in diameter (5 individuals in each hole), drilled on opposite sides on the upper part of each host fruit. Implanting first instar larvae instead of eggs assured that no infertile or dead eggs were used. Subsequently, infested fruits were individually placed in plastic containers on a layer of sterilized sand, were covered with organdie cloth and transferred in three different rooms, adjusted to 45–55% RH and i) 15˚C, ii) 25˚C, and iii) 30˚C. On a daily basis, all artificially infested fruits were carefully inspected and newly formed pupae were collected. We performed 20 replications for each host fruit (apples, bitter oranges), medfly population

(Benakeio, 88.6, S10.3 and wildish) and temperature (15˚C, 25˚C, 30˚C). Hence, we used 240 apples and 240 bitter oranges, in total (20 for each of the 4 medfly populations and the 3 temperatures). Also, the total number of first instar larvae implanted in fruit was 4800 (2400 in apples and 2400 in bitter ornages).

### Data analysis

The effects of *Wolbachia* infection, fruit-host, and temperature on survival during immature development were tested with the binary logistic regression. Binary logistic tests the effect of several factors and their interactions on a dichotomous depended variable (i.e. survival; dead-alive). Wald *t*-test or z value was used to assess the significance of the tested predictors. We also used the Cox proportional hazards model to assess the effects of the above factors and of sex on the duration of larval, pupal and total immature development. This survival model is commonly used in medical and demographic research to assess the association between time to event (i.e. time to pupation/adult emergence) and one or more predictors. More specific, the model allowed examining the effects of the *Wolbachia* infection, fruit-host, and temperature on larval, pupal and larva to adult developmental duration [69]. All data analyses were conducted using the statistical software SPSS 25.0 (SPSS Inc., Chicago, IL, U.S.A.).

## Results

### Survival rates

The survival rates of the immature stages of all four medfly populations, in both fruit species (apples, bitter oranges) held under three constant temperatures (15, 25, 30˚C) are given in Fig 1. Overall larva-to-pupa and larva-to-adult survival followed similar patterns and varied a lot among the four medfly populations, the two host fruits and the different temperatures (Fig 1A and 1C). Pupation rates were higher for larvae implanted in apples compared to bitter oranges regardless of the *C. capitata* population and the temperature (Wald test $t = 87.07$, df = 1, $P < 0.001$). Medfly population was also a significant predictor of larva-to-pupa survival (Wald test $t = 144.4$, df = 3, $P < 0.001$). The performance of wildish ($F_9$) medflies was higher than that of the laboratory adapted ones, particularly in bitter oranges. The *Wolbachia* infected medfly populations, expressed lower larva-to-pupa survival rates, especially the *w*Cer4 infected one (S10.3). Temperature differentially affected larva-to-pupa survival rates (Wald test $t = 47.24$, df = 2, $P < 0.001$). Higher temperature decreased larva-to-pupa survival in wildish, Benakeio and 88.6 populations when larvae developed in apples, but did not affect it when larvae developed in bitter oranges (especially in wildish flies). Exposure to 30˚C was detrimental for the laboratory adapted flies and lethal for the larvae of *w*Cer4 *Wolbachia* infected population S10.3.

The interaction between fruit type and medfly population was significant (Wald test $t = 63.32$, df = 3, $P < 0.001$). Indeed, larva-to-pupa survival rates of Benakeio and the *w*Cer2 infected population (88.6) were higher in apples compared to bitter oranges, while the wildish and the S10.3 flies expressed similar survival rates in the two host fruits (Fig 1A). Likewise, the interaction between fruit and temperature was significant (Wald test $t = 31.16$, df = 2, $P < 0.001$) since survival rates varied a lot among different temperatures in apples while they were more stable in bitter oranges. The significant interaction between medfly population and temperature (Wald test $t = 47.82$, df = 6, $P < 0.001$) highlights the dramatic drop in survival rates of the laboratory adapted flies (Benakeio, 88.6, S10.3) compared to wildish ones in higher temperatures.

The survival rates at pupal stage for the four medfly populations, the two hosts (apple, bitter orange) and the three temperatures is given in Fig 1B. Host fruit was a significant predictor of

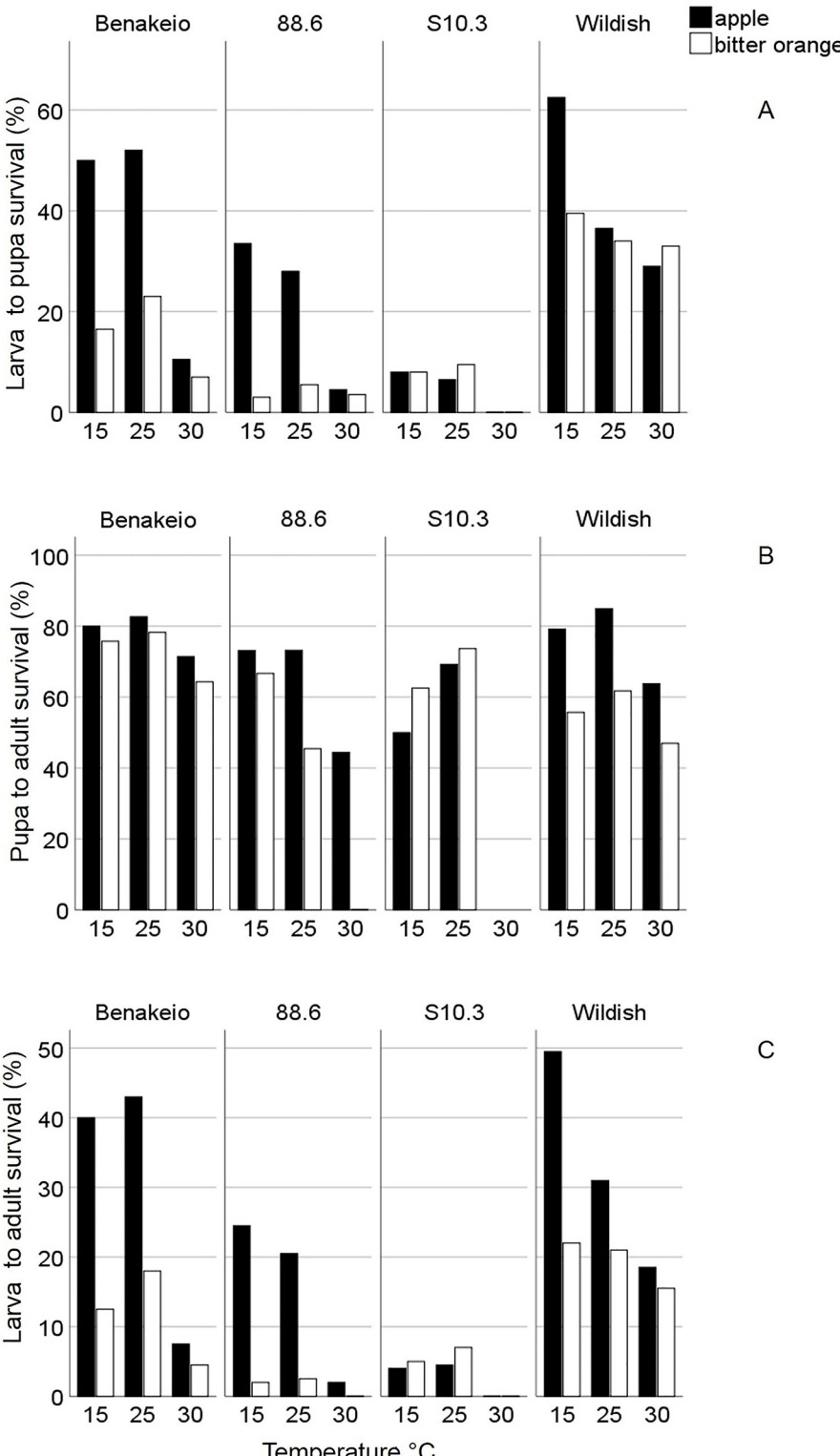

**Fig 1.** Survival rates of larva-to-pupa (A), pupa-to-adult (B) and larva-to-adult (C) of i) *Wolbachia* uninfected laboratory medfly population "Benakeio", ii) *w*Cer2 infected medfly population "88.6", iii) *w*Cer4 infected medfly population "S10.3" and iv) wildish (uninfected) medfly population collected in the region "Agia", in two different hosts (apples and bitter oranges).

pupae survival rates regardless of medfly population and temperature (Wald test $t = 7.70$, df = 1, $P = 0.006$) with pupae obtained from apples surviving at higher rates compared to those obtained from bitter oranges. Medfly population was a significant predictor as well (Wald test $t = 11.15$, df = 3, $P = 0.011$), as pupae of the Benakeio population survived at higher rates compared to the *Wolbachia* infected and the wildish ones, regardless of the host fruit and temperature. Lower temperatures tended to increase the pupae survival rates of all medfly populations in both host-fruit (Wald test $t = 17.78$, df = 2, $P < 0.001$). The interaction between host-fruit and medfly population was significant (Wald test $t = 10.75$, df = 3, $P = 0.013$) indicating differential performance of the four populations in the different fruits. Pupae of the Benakeio, derived either from bitter oranges or apples, expressed similar survival rates. However, pupae of the wildish and the 88.6 populations that derived from apples survived at higher rates compared to those obtained from bitter oranges. On the contrary, *w*Cer4 infected pupae (S10.3) reared in bitter oranges survived at higher rates than those reared in apples.

Larva to adult survival rates (percentage of adult emergence) of all four medfly populations, reared in both apples, and bitter oranges held under 15, 25, and 30°C are given in Fig 1C. Similar to larva-to-pupa survival rates, the host had a significant effect on adult emergence rates regardless of the population and the rearing temperature (Wald test $t = 57.40$, df = 1, $P < 0.001$). The percentage of adults that survived in apples was much higher than in bitter oranges, with the exception of S10.3. Medfly population also significantly affected adult survival rate (Wald test $t = 103.46$, df = 3, $P < 0.001$) as a higher percentage of adults emerged from the Benakeio and wildish populations compared to 88.6 and S10.3. Likewise, rearing temperature was a significant predictor of adult emergence rates regardless of the host and *C. capitata* population (Wald test $t = 30.44$, df = 2, $P < 0.001$). Overall, fewer adults emerged in higher temperatures as a proportion of the total larvae implanted into fruit.

The interaction between fruit type and medfly population was significant (Wald test $t = 42.04$, df = 3, $P < 0.001$). In detail, adult survival rates of the Benakeio, the *w*Cer2 infected population (88.6) and the wildish flies were higher in apples compared to bitter oranges while the S10.3 expressed almost similar survival rates in the two host fruits. The interaction between fruit and temperature was significant as well (Wald test $t = 12.69$, df = 2, $P = 0.002$). Survival rates were higher at low temperatures in apples, but did not reveal a clear trend at different temperatures in bitter oranges. A significant interaction between medfly population and temperature was also recorded (Wald test $t = 32.15$, df = 6, $P < 0.001$). The survival rates of the laboratory adapted flies (Benakeio, 88.6, S10.3) were much lower compared to wildish ones in higher temperatures.

## Developmental duration

The immature stages developmental duration (days) of the four medfly populations, in both apples, and bitter oranges, held under 15, 25 and 30°C is given in Fig 2 and Table 1. The larval development duration was longer in apples compared to bitter oranges regardless of the *C. capitata* population and the rearing temperature (Wald test $t = 17.29$, df = 1, $P < 0.001$) (Fig 2A). Medfly population was also a significant predictor of larval developmental duration regardless of the host-fruit and the rearing temperature (Wald test $t = 40.48$, df = 3, $P < 0.001$). In particular, larval developmental duration of the S10.3 (*w*Cer4 infected line) was the longest among the four populations tested. As expected, the temperature affected significantly the larval developmental duration regardless of the host-fruit and medfly population (Wald test $t = 847.20$, df = 2, $P < 0.001$), with longer duration reported at 15°C. The only significant interaction recorded was that between host-fruit and *C. capitata* population (Wald test $t = 10.5$, df = 3, $P = 0.015$), indicating smaller differences in larval developmental duration between

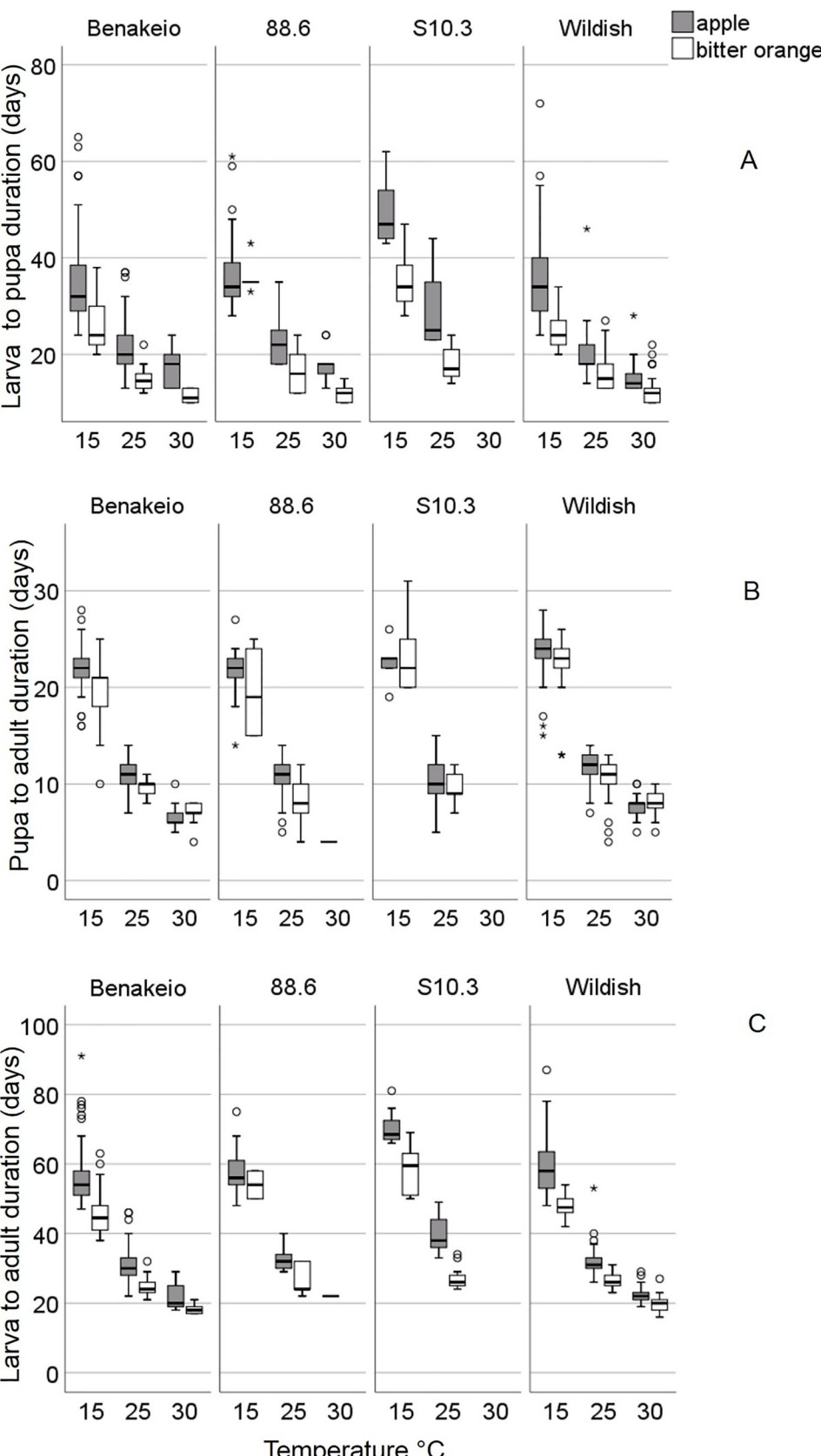

**Fig 2.** Immature developmental duration of larva-to-pupa (A), pupa-to-adult (B) and larva-to-adult (C) of i) *Wolbachia* uninfected laboratory medfly population "Benakeio", ii) *w*Cer2 infected medfly population "88.6", iii) *w*Cer4 infected medfly population "S10.3" and iv) wildish (uninfected) medfly population collected in the region "Agia", in two different hosts (apples and bitter oranges).

apples and bitter oranges in the Benakeio, the wildish and the 88.6 populations compared to the S10.3.

Pupal developmental duration of the four medfly populations, in both apples and bitter oranges held under 15, 25, and 30˚C are given in Fig 2B. Host-fruit was a significant predictor of pupal developmental duration regardless of the medfly population and the temperature (Wald test $t = 4.25$, df = 1, $P = 0.039$). Adults emerged later from pupae reared in apples than in bitter oranges. The effect of *C. capitata* population on the developmental duration of pupae was significant regardless of the host-fruit and the temperature (Wald test $t = 82.59$, df = 3, $P < 0.001$). Temperature significantly affected the developmental duration of pupae as well (Wald test $t = 394.35$, df = 2, $P < 0.001$). As in the case of larvae development, pupal developmental rates were significantly slower in low temperatures (15˚C). Finally, the interaction between host-fruit and temperature was significant (Wald test $t = 118.03$, df = 2, $P < 0.001$) indicating the differential effect of temperature on pupal developmental duration that derived from different host-fruits.

Host-fruit, medfly population and temperature were significant predictors of larva-to-adult developmental duration (Wald test $t = 260.98$, 49.71 and 915.38; df = 1, 3 and 2 $P < 0.001$, respectively) (Fig 2C). Alike larval and pupal developmental duration, total larva-to-adult developmental duration was longer in apples than in bitter oranges regardless of the medfly population and the temperature. Total immature developmental time was significantly shorter in high temperature, although the *Wolbachia* infected medfly population 88.6 and S10.3 largely failed to complete development at 30˚C. The *w*Cer4 infected medfly population (S10.3) exhibited the longest total developmental duration compared to the three other medfly populations.

In a separate Cox regression analysis, testing whether the sex (male, female) is a significant predictor of the i) pupa-to-adult, and ii) larva-to-adult developmental duration we found no significant effects (Wald test t = 0.22 and 1,21; df = 1; P = 0.64 and 0,27 respectively) (Fig 3).

**Table 1. Variables of Cox proportional hazards model for the developmental duration of *C. capitata* immature stages.**

| stage | r | B | SE | Exp (B) | P |
|---|---|---|---|---|---|
| larva-to-pupa | fruit | 0.93 | 0.22 | 2.53 | <0.001 |
| | population | | | | <0.001 |
| | temperature | | | | <0.001 |
| | fruit*population | | | | 0.015 |
| pupa-to-adult | fruit | 0.27 | 0.13 | 1.31 | 0.039 |
| | population | | | | <0.001 |
| | temperature | | | | <0.001 |
| | sex | -0.04 | 0.08 | 0.97 | 0.64 |
| | fruit*temperature | 0.44 | | | <0.001 |
| larva-to-adult | fruit | 1.51 | 0.09 | 4.52 | <0.001 |
| | population | | | | <0.001 |
| | temperature | | | | <0.001 |
| | sex | -0.09 | 0.08 | 0.92 | 0.27 |

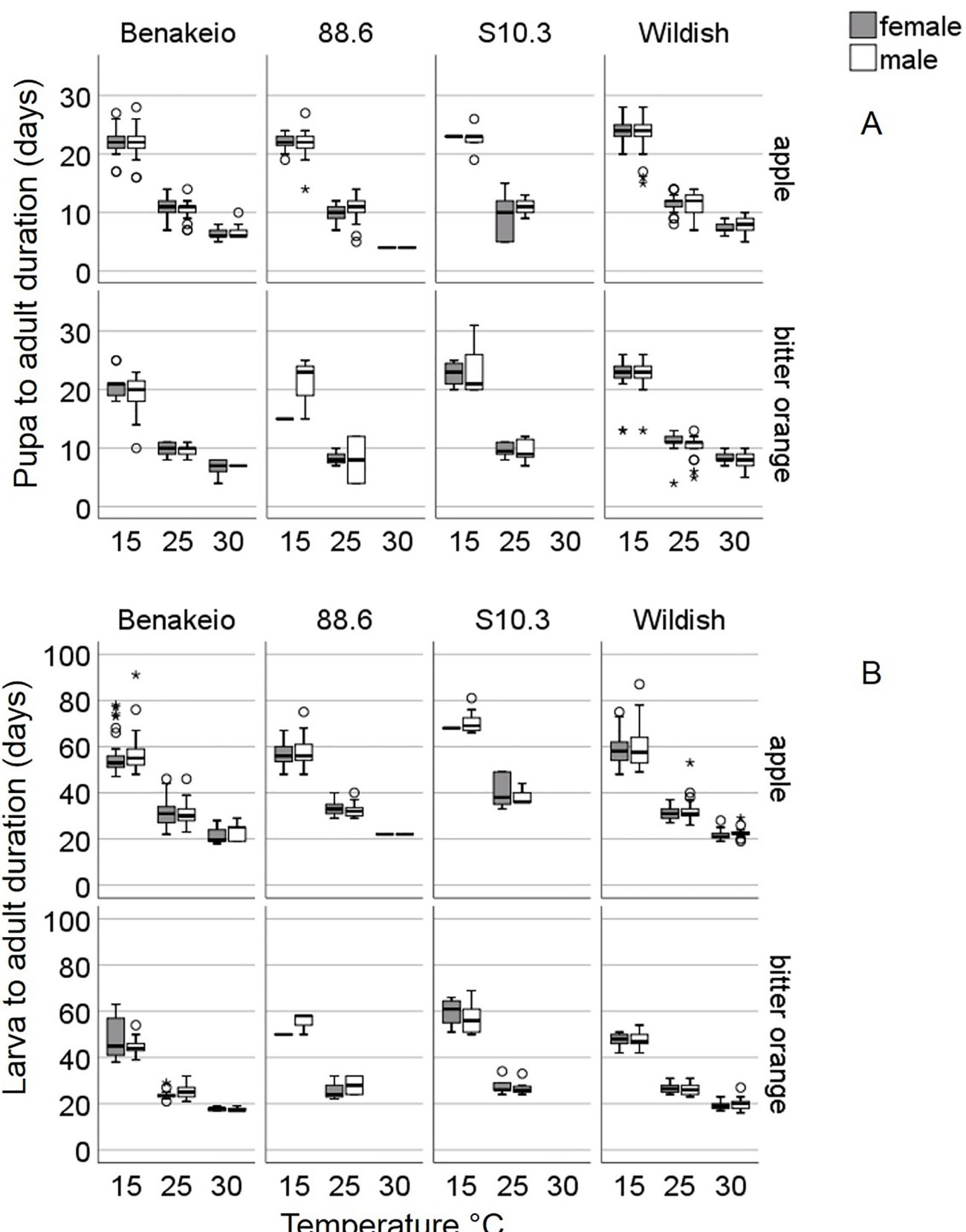

**Fig 3.** Developmental duration of pupa-to-adult (A) and larva-to-adult (B) for males and females obtained from i) *Wolbachia* uninfected laboratory medfly population "Benakeio", ii) *w*Cer2 infected medfly population "88.6", iii) *w*Cer4 infected medfly population "S10.3" and iv) wildish (uninfected) medfly population collected in the region "Agia", in two different hosts (apples and bitter oranges).

## Discussion

### Effect of laboratory adaptation and *Wolbachia* infection on survival

The results of the current study demonstrate that wildish flies perform better in different host fruits and temperatures compared to laboratory adapted ones that are either trans-infected (*w*Cer2, *w*Cer4) or not infected with *Wolbachia*. This is strongly supported by the small differences between the two hosts and among the three temperatures regarding survival rates of the immatures of wildish flies compared to the other three populations tested. It is well documented that during domestication, fruit flies and other insects go through major bottlenecks that reduce genetic variability. Adaptation to artificial diet and artificial oviposition devices are the two main selection forces operating, with the first one being more important [70,71]. Loss of genetic variability of captive populations of fruit flies (i.e. in mass rearing facilities) results in commonly observed changes such as reduced developmental time, earlier reproduction, increased fecundity, as well as reduced lifespan, dispersal ability and stress resistance [72–75]. Increased mortality of *Wolbachia* infected laboratory populations may indicate an additional bottleneck effect during the trans-infection of the maternal line. Introduction of wild individuals or crosses of inbred lines are required to sustain behavioral integrity and competitive ability of mass reared insect populations [76,77]. In our experiments, intensive artificial rearing in laboratory conditions may have resulted in adaptive and plastic changes, such as loss of specific alleles that contribute to successful and efficient development in wild hosts. This is strongly supported by previous studies indicating that laboratory adapted populations of the Mediterranean fruit fly perform better in artificial rearing medium (% of egg hatch, larval and pupal survival) than wild populations [78].

The Mediterranean fruit fly exhibits an impressive ability to conclude development in a long list of hosts including fruit species containing secondary metabolites that are detrimental in other insect species [79]. Medfly larvae can also successfully develop in fruit species that are considered nutritionally poor such as cotton bolls [80]. Larvae of laboratory adapted populations are maintained at high nutritious diets that contain no secondary plant metabolites. Adaptation to these diets may result in loss of the ability to efficiently acquire nutrients and to cope with defense compounds of fruit hosts such as apples and bitter oranges. In general, apples are considered not a preferable host for *C. capitata* larvae containing low amounts of nitrogen and high of phenolic compounds [81–83]. On the other hand, many citrus fruits contain high quantities of essential oils in the flavedo area that impede survival of young larvae [44,84]. Although bitter oranges are among the favored hosts for the Mediterranean fruit fly compared to apples, our results demonstrate higher survival rates of immatures in the latter, especially for the laboratory adapted populations. Nash and Chapman [85] showed that each developmental stage of medfly responded differently to alterations in specific dietary nutrients. For example, the mortality and the developmental rates of medfly larvae increased when dietary protein quality and availability decreased, but mortality in pupal stage increased with alteration of carbohydrate quality.

Laboratory insect populations live generally under more preferable conditions than wild populations (optimal temperature, constant availability of mates and food, absence of predators). These relaxed conditions lead to lack of selection pressures that adverse environments infer to wild populations and to low expression of the stress-related genes (such as heat shock

proteins). As adaptation of organisms to new environments is driven by genes that control generalized stress resistance, loss of stress resistance in laboratory rearing conditions has been documented before [74,86,87]. The results of the current study are aligned with those mentioned earlier. Laboratory adapted flies reared under benign conditions for long periods expressed lower performance compared to wildish flies when subjected to rather moderate temperature and host fruit stressful conditions.

Besides generic factors, long-lasting artificial rearing using larval diets that often contain preservatives and antibiotics results in alteration of both diversity and abundance of the fly's microbiome [66]. Gut bacteria are important determinants of tephritid (and other insect species) fitness affecting a range of biological traits such as immatures development and survival, adult reproduction and longevity, as well as sexual performance and chemosensory responses [88–90]. Nitrogen fixation seems to contribute to adult performance while both bacterial regulated nitrogen and carbon metabolism may affect larvae development and fitness [90,91]. Host fruit is a central point of medfly–gut bacteria interaction that involves the spread of bacteria through vertical transmission and the fruit metabolism that enhances nutrient acquisition by larvae. Destruction of the bacterial community in laboratory adapted flies may be one of the factors contributing to reduced larvae performance of the laboratory adapted flies to wild hosts. Wildish flies may have also lost part of the symbiont fauna during rearing for 9 generations in laboratory conditions before being used in our experiments. However, the small number of generations in captivity may not be detrimental for important facultative and obligate gut bacterial communities. Apparently, a comparative study of the structure and diversity of the symbiont communities in populations tested is required to elucidate the above explanation.

## Effect of temperature on survival

Within the range of tested temperatures (15–30˚C) the lower the temperature the higher the survival rates of medfly immatures, with laboratory adapted populations, especially the *Wolbachia* infected ones, being more sensitive to temperature rise. In general, lower temperatures promote immature survival rates [38–41,92] and longer developmental duration, providing additional time to acquire important nutrients from poor diets (such as apples), considering also possible absence of important elements of the symbiotic fauna. Also, longer duration may allow for detoxifying secondary metabolites that are detrimental under a "fast track" developmental duration. Lower temperatures seem to be more beneficial for flies that are not well adapted to wild hosts. Indeed, the wildish population performed much better in both host fruits at 25 and 30˚C compared to infected and non-infected Benakeio flies. Loss of genetic diversity and destruction of the symbiotic fauna may also account for the significant interaction between medfly population and temperature. Temperatures higher than 25˚C result in minimal survival of the *Wolbachia* infected flies compared to the non-infected ones, probably indicating a microbiome deficiency and/or lack of genetic diversity. In fact, survival was below 1% for the immatures of the *w*Cer4 infected line. In general, *Wolbachia* is sensitive to higher temperatures and an increase in temperature is expected to result in lower titers [93,94], and smaller effects on insect hosts in contrast to our findings. A genetic/genomic comparison to test the genetic variability hypothesis could be addressed in future studies.

## Effect of host fruit on survival

For all four medfly populations tested, the two host fruits and the three temperatures, survival rates increased dramatically after pupation. It seems that pupation is a milestone in the development of holometabolous insects and tephritids more specifically [41,45,95]. Pupae obtained

from apples survive at higher rates compared to those from bitter oranges. Host fruit even within plant genus and larval diets have been found to significantly affect pupal survival in medflies [44,45,81,95]. Secondary metabolites and nutritional factors may account for the reduced survival of pupae obtained from bitter oranges. Longer developmental duration of larvae in apples compared to bitter oranges that promotes accumulation of nutritional compounds may be related with the subsequent higher pupal survival rates. However, other studies conducted under optimal temperature conditions have not demonstrated positive correlation between larval developmental duration and survival with pupal performance [10,44,45]. The physicochemical characteristics of host fruit (such as pH and soluble solid contents) may affect the performance of *C. capitata* immature stages [96].

## Effect of host fruit and temperature on developmental duration

Similar to other earlier studies, overall larval developmental duration was longer in apples compared to bitter oranges [10,44]. Host fruit nutritional and physicochemical characteristics exert strong effects in larval developmental duration of medfly [85,97]. Carey [98] reported that larval development of *C. capitata* increased from 1 week in favorable hosts such as mango and tomato to more than 3 weeks in quinces. Host fruit cultivar, and the rate of ripening can also affect the developmental duration of *C. capitata* [10,99]. Kaspi et al. [100] reported that medfly larval development in protein-rich diets was faster and that emerging adults were larger. The protein levels in fruit hosts such as fig, peach and orange increase during ripening favoring shorter larval development. Apart from nutritional elements, the host's secondary metabolites and the firmness of the mesocarp may affect the larval developmental time. In our experiments, high larval mortality and long developmental period indicate that apple may not be one of the favorable hosts for medfly, and this is supported by an older study suggesting that the flesh density in apples inhibits the development of young larvae [101]. Larval developmental duration longer than five months (from October to April) in field-maintained apples has been recorded in the area of Thessaloniki northern, Greece.

As it was expected lower temperatures increased developmental duration of immatures [102]. Extension of the larval stage under low temperatures in key hosts such as apples is important for the overwintering of *C. capitata* in cooler marginal for its establishment areas [8]. Our data reveal much larger extension of the duration of the immature stages in apples compared to bitter oranges highlighting the importance of the host fruit for the overwintering of medfly in cooler areas. Wildish flies exhibit higher plasticity in larval developmental duration compared to non-infected Benakeio flies. Factors such as loss of generic diversity and symbionts during long periods of domestication may account for the observed difference (see discussion above). Pupal developmental duration was affected by larvae food and medfly population as it has been demonstrated in other studies, as well [44,45,85].

## Effect of *Wolbachia* on developmental duration

Our study demonstrates for very first time the demographic alterations imposed by *Wolbachia* when medfly immature development takes place on natural host fruits. *Wolbachia* infected medflies, expressed the lower survival rates and the higher developmental times. Moreover, *Wolbachia* infection seems to exert differential responses on medfly survival and development in different fruit hosts. The *w*Cer4 infected medflies showed remarkably low survival rates in both apples and bitter oranges, whereas the *w*Cer2 infected ones were less vulnerable when developed in apples compared to bitter oranges. Also, immature development seems to be more prolonged in apples compared to bitter oranges. Despite the lack of previous data on the effects of *Wolbachia* infection on medfly demography when immature rearing occurred in

fruit hosts, a series of studies have recorded the symbiosis-imposed alterations in artificial larval diet [62,64,103]. *Wolbachia* infection increases immature mortality in medfly, mainly during the egg and larval stage [64]. Negative effects of *Wolbachia* infection on pupae are much smaller. Differential effects of the two *Wolbachia* strains have also been reported earlier, with more negative effects imposed by *w*Cer4 strain. In addition, a female fertility advantage of *w*Cer2 over *w*Cer4 has also been mentioned previously [62,64].

In general, the significant role of microbiota (*Wolbachia*, gut bacteria etc.) in tephritid biology and behavior is widely accepted [65,104]. However, the interaction among the different bacterial communities and the way that these interactions affect insect's biology and behavior are not completely unraveled. Recently, Simhadri et al. [105] showed that *Wolbachia* could modify the gut microbiome in *Drosophila melanogaster*. This finding points out the importance of studying insects as a "holobiont" where the bacterial interactions determine important biological traits. In addition, abiotic environment (e.g. fruit host, temperature) could also interact with "holobionts" and largely determine their bacterial community (e.g. quantitative and qualitative composition of gut microbiota, *Wolbachia* titer etc.) Hence, the biological traits exerted by a specific *Wolbachia* strain in a given fruit host should be attributed more to the symbiotic "interactions" than to the specific attributes recognized either for the fruit host or the *Wolbachia* strain when studied separately. In fact, medfly-*Wolbachia*-fruit interactions could explain the differential responses on medfly survival and development in different fruit hosts that were observed in our study. Obviously, future research that would shed light on the insects' bacterial interactions is essential in order to understand the "holobiont's" biological function.

## Supporting information

**S1 Dataset. Dataset.xls file with raw data.**
(XLSX)

## Author Contributions

**Conceptualization:** Nikos T. Papadopoulos.

**Data curation:** Niki K. Dionysopoulou.

**Formal analysis:** Niki K. Dionysopoulou, Stella A. Papanastasiou.

**Funding acquisition:** Nikos T. Papadopoulos.

**Investigation:** Niki K. Dionysopoulou.

**Project administration:** Nikos T. Papadopoulos.

**Resources:** Nikos T. Papadopoulos.

**Supervision:** Nikos T. Papadopoulos.

**Validation:** Stella A. Papanastasiou.

**Writing – original draft:** Niki K. Dionysopoulou, Nikos T. Papadopoulos.

**Writing – review & editing:** Stella A. Papanastasiou, Georgios A. Kyritsis, Nikos T. Papadopoulos.

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
