## [Decision Letter · Decision Letter 0]

7 Nov 2019

PONE-D-19-25021

Effect of host fruit, temperature and Wolbachia infection on survival and development of Ceratitis capitata immature stages

PLOS ONE

Dear Dr Papadopoulos,

Thank you for submitting your manuscript to PLOS ONE. After careful consideration, we feel that it has merit but does not fully meet PLOS ONE’s publication criteria as it currently stands. Therefore, we invite you to submit a revised version of the manuscript that addresses the points raised during the review process.

Please try to carefully study and responde all the queries raised by both reviewers before you attempt to return your revised manuscript for further revision. Be sure to carefully check the grammar and also make the text more clear, based on your proposed objectives.

We would appreciate receiving your revised manuscript by Dec 22 2019 11:59PM. To enhance the reproducibility of your results, we recommend that if applicable you deposit your laboratory protocols in protocols.io, where a protocol can be assigned its own identifier (DOI) such that it can be cited independently in the future. For instructions see: http://journals.plos.org/plosone/s/submission-guidelines#loc-laboratory-protocols

We look forward to receiving your revised manuscript.

Kind regards,

Luciano Andrade Moreira, PhD

Academic Editor

PLOS ONE

Journal Requirements:

1. We noticed you have some minor occurrence of overlapping text with the following previous publication(s), which needs to be addressed:

https://academic.oup.com/aesa/article/95/5/564/28680

In your revision ensure you cite all your sources (including your own works), and quote or rephrase any duplicated text outside the methods section. Further consideration is dependent on these concerns being addressed.

Reviewers' comments:

Reviewer's Responses to Questions

**Comments to the Author**

1. Is the manuscript technically sound, and do the data support the conclusions?

Reviewer #1: Yes

Reviewer #2: Yes

2. Has the statistical analysis been performed appropriately and rigorously? 

Reviewer #1: Yes

Reviewer #2: Yes

3. Have the authors made all data underlying the findings in their manuscript fully available?

Reviewer #1: Yes

Reviewer #2: Yes

4. Is the manuscript presented in an intelligible fashion and written in standard English?

Reviewer #1: No

Reviewer #2: Yes

5. Review Comments to the Author

Reviewer #1: The manuscript is technically sound, and the statistical analysis performed appropriately and rigorously. The authors have made all data available.

Overall the information provided in the manuscript is new and interesting. However, the manuscript needs to be edited for grammar. I also found the manuscript too long with the information buried on how Wolbachia infection affected the different fruit fly lines and how the research could be applied. The section on Wolbachia in the introduction should be moved to under the introductory paragraph, and the manuscript also needs to be edited carefully as some sentences did not make sense. For example, lines 62, 82, 83. Also note that it should be Wiedemann NOT Wiedeman.

Reviewer #2: Comments to the Authors:

The authors test the influence of two different host fruits, three different constant temperatures and the population effects of three laboratory lines (Wolbachia-infected and -uninfected) and one semi-wild population on developmental times and survival of larval and pupal stages of Ceratitis capitata. I congratulate the authors for their efforts in this experimental work.

The authors present convincing evidence of a significant influence of each of these variables (and their interaction) on developmental time and survival of C. capitata. The most interesting finding, however, is the influence of Wolbachia on differential survival and performance in the two tested host fruits. The effects of Wolbachia on survival and development of C. capitata in relation to host fruit and temperature is a new finding in this system and brings novelty to the field.

I have minor issues with the English used in this study. I think that some sentences are not clear and a little more elaboration is needed at certain points throughout the manuscript. As there weren’t many writing mistakes, I have taken the time to point some of them out. I still recommend a careful revision of the manuscript. Below, find my comments to the authors.

Minor comments:

>L19: Include the Latin name of species.

>L40: Remove “the” two times. Occurs throughout the manuscript.

e.g. “... expressed lower survival rates and higher developmental times...”

>L44: Not sure I see how “Practical and theoretical implications” are discussed in the discussions. Instead, it appears that implications on the ecology and survival of the fly in nature are discussed.

>L59: Change word “other” to “others”.

e.g. “... some are favourable, while others, are marginal...”

>L60: I think the implications apples have on the over-wintering success of this species in temperate zones is interesting. Perhaps a few more words explaining this could be provided in the introduction.

>L73: Remove parentheses and make meaning clearer. What I think was meant:

“... immatures after being exposed to artificially infested fruit at low storage temperatures for ...”

>L76: I am not sure if the use of the word “biotypes” is appropriate here. In the given literature the word “biotypes” is not mentioned. I would use the word “populations” or “flies” instead. If the word “biotypes” is used, please include an explanation of what is meant by it.

>L78: The term “wildish populations” sounds a little odd. Please make clear that these are populations from the wild that have been reared in laboratory conditions for a certain amount of generations.

>L82: Replace “to an” with “in an”.

e.g. “... this process results in an increased ...”.

>L83: Remove “the”. Replace “on” with “in” and “to a” with “in”.

e.g. “... fitness under artificial rearing conditions and in the same time in reduced performance under different rearing conditions ...”.

I think this sentence, in general, should be double checked.

>L86: Include “as”.

“... factors such as temperature ...”

>L87: Again, the use of “biotypes”. See above (e.g. different medfly lines, populations, etc...).

>L91: Replace “a-proteobacteria” with “α-proteobacteria”.

>L103-109: In this part of the introduction I think it is important to mention how the kind of fitness alterations caused by Wolbachia onto its hosts can be dependent on how long the host is infected with Wolbachia. For example, new or recent infections more commonly result in higher fitness costs than long-standing ones. This will emphasize some differences between natural infections and transinfected lines, where Wolbachia was transferred from a donor to recipient recently. Otherwise, I agree that the relationship between Wolbachia and host represents a dynamic nature not simply explained by any one factor alone and warrants investigation of interactions with abiotic factors.

>L125: Citation “58” is wrongly placed here. Riegler & Stauffer, 2002 did not do laboratory crossings to establish level of CI. They analysed natural populations for infection frequencies. Plus, it is not clear if the authors refer to R. cerasi of C. capitata here.

>L147: Although the term “wildish” is explained in the Materials and methods section, I think it would be good to have a few additional words that explain what is meant by it already in the abstract or introduction. I may be wrong, but the term sounds a little awkward without further description.

>L148-152: Sentence should be re-stated. Was it addressed or wasn’t it addressed?

e.g. “By including Wolbachia-infected lines in our experiments, we addressed whether infection affects ...”

>L158: For reproducibility reasons, it may be worth noting what kind of lights were used (I’d assume this is more important for the adult rearing conditions).

>L161: Italicise the “w” in “wCer2”.

>L171: “... for few months ...” May the authors provide a range or approximation of how many months in more exact terms?

>L176: Closing parentheses missing.

>L194: I think ImageJ may require a citation. Additionally, what was done with these measurements (length and width of pupae)? In the results, it appears that only survival and developmental times were recorded, so it is a little unclear to me what was done with these measurements and how they were incorporated into this study?

> In the supplemental excel file, it is not clear to me how many samples and which samples belong to which of the 20 replicates (L195). This could be clearer. Maybe just by adding a total of how many individuals were finally used.

>L199-204: Although I think the statistical procedures used in this study are sound, I would appreciate to read why they were chosen. Perhaps advantages, disadvantages, or even just references and what exactly was done using them (e.g. tests of significance, interactions, etc...). Also applies to the Wald test, which is only mentioned in the “Results” section (this could be included in the “Materials and methods” section, stating that it was used in the regression analyses to test for significance).

> I recommend adding sub-headers for each section within results and discussions for readability (maybe even materials and methods). E.g. survival rates, performance/developmental duration, etc... Otherwise, the results are presented clearly. The figures and tables are also well made and clear.

>L217: Remove “the”.

“... expressed lower egg-to-pupa survival rates ...”

>L344-348: While I agree that the comparison of wildish flies to the non-infected Benakeio laboratory medfly line supports the idea that reduced genetic variation and an altered microbiome in laboratory lines may result in poorer survival and performance on host fruit, I think it is hard to rule out that the poorer performance of the 88.6 and S10.3 lines is due to the same reasons. Perhaps an additional bottleneck effect might have occurred while trans-infecting these lines and may be worth acknowledging that Wolbachia may not be the sole reason these lines under-perform, especially under higher temperatures where the influence of Wolbachia is supposed to be hampered.

>L438-441: In accordance with the previous comment, if higher temperatures may be detrimental for Wolbachia infections and their influence on hosts, the fact that higher temperatures in this study affected the infected lines most may be indicative of microbiome deficiency or lack of genetic diversity, which the authors include. The authors mention how the microbiome hypothesis could be tested (L421). A genetic/genomic comparison to test the genetic variability hypothesis may be mentioned as well.

>L472-476: These two sentences are not very clear. I’ve missed the take home message of Krainacker et al. Also, next sentence, I think it needs to be re-written for clarity.

> I recommend a thorough inspection of the references section as some inconsistencies and mistakes seem to be present.

Here are just a few examples:

>L611: Reference is missing journal name, volume and page numbers.

>L738: Reference is missing page number.

>L752: Journal name format is different from others.

>L766: Journal name format is different from others.

>L775: Journal name format is different from others.

>L786: Journal name format is different from others.

6. PLOS authors have the option to publish the peer review history of their article (what does this mean?). If published, this will include your full peer review and any attached files.

Reviewer #1: No

Reviewer #2: Yes: Vid Bakovic

---

## [Author Response · Author response to Decision Letter 0]

20 Dec 2019

To 

Luciano Andrade Moreira, PhD

Academic Editor

PLOS ONE 

Dear Editor

With reference to your e-mail of November 8th I am returning to you our revised manuscript “Effect of host fruit, temperature and Wolbachia infection on survival and development of Ceratitis capitata immature stages” (Manuscript ID PONE-D-19-25021). 

We considered the points raised by your behalf as well as by the two reviewers and we related to all of them. English language has been improved throughout the text and our arguments are now made clearer. Overlapping text with a previous publication of our group is now rephrased and all literature sources are cited. Captions of the supporting information files are included at the end of the manuscript. Moreover, in the “Materials and Methods” section the description of the “egg implanting” procedure in fruit has been corrected to “first instar larvae implanting” in fruit. The whole procedure is now described in detail. Phrases such as “egg to pupa survival/development” etc. have been modified in “larva to pupa survival/development” etc. in Figures, Figure legends, Tables and text. A list of responses to each of the Reviewers’ major comments is given bellow: 

Reviewer #1

General remarks:

1. “….the manuscript needs to be edited for grammar.”

Answer: We agree with reviewer’s recommendation and an effort has been made to improve the English language throughout the text. 

2. “I also found the manuscript too long with the information buried on how Wolbachia infection affected the different fruit fly lines and how the research could be applied.”

Answer: An effort has been made to reduce the text (see deleted parts). Also, subheadings are now included in the “Results” and “Discussion” sections to give emphasis and improve readability of the manuscript.

3. “The section on Wolbachia in the introduction should be moved to under the introductory paragraph.”

Answer: Done

4. “…the manuscript also needs to be edited carefully as some sentences did not make sense. For example, lines 62, 82, 83.”

Answer: An effort has been made to improve the language throughout the text. Also, we have rephrased the sentences in lines 62, 82 and 83 making the context clearer (lines 65-66, 112-114). 

5. “Also note that it should be Wiedemann NOT Wiedeman.”

Answer: Corrected (line 54)

Reviewer #2

General remark: “I have minor issues with the English used in this study. I think that some sentences are not clear and a little more elaboration is needed at certain points throughout the manuscript. I recommend a careful revision of the manuscript.”

Answer: We agree with reviewer’s recommendation and an effort has been made to improve the English language throughout the text.

Specific remarks: (numbering follows the list of comments by Reviewer #2):

1. Line 19: “Include the Latin name of species”

Answer: Done (line 19).

2. Line 40: “Remove “the” two times. Occurs throughout the manuscript. e.g. “... expressed lower survival rates and higher developmental times...””

Answer: The sentence has been changed to “... expressed lower survival rates and higher developmental times...” (line 40).

3. Line 44: “Not sure I see how “Practical and theoretical implications” are discussed in the discussions. Instead, it appears that implications on the ecology and survival of the fly in nature are discussed”

Answer: The phrase has been changed to “Implications on the ecology and survival of the fly in nature are discussed” (lines 44-45).

4. Line 59: “Change word “other” to “others”. e.g. “... some are favourable, while others, are marginal...””

Answer: Corrected (line 59). 

5. Line 60: “I think the implications apples have on the over-wintering success of this species in temperate zones is interesting. Perhaps a few more words explaining this could be provided in the introduction.”

Answer: The implications of apple infestation on the successful overwintering of medfly are now elaborated in lines 60-64. 

6. Line 73: “Remove parentheses and make meaning clearer. What I think was meant: “... immatures after being exposed to artificially infested fruit at low storage temperatures for ...””

Answer: Clarified (lines 104-105). 

7. Line 76: “I am not sure if the use of the word “biotypes” is appropriate here. In the given literature the word “biotypes” is not mentioned. I would use the word “populations” or “flies” instead. If the word “biotypes” is used, please include an explanation of what is meant by it.”

Answer: Corrected in line 107 and throughout the text. 

8. Line 78: “The term “wildish populations” sounds a little odd. Please make clear that these are populations from the wild that have been reared in laboratory conditions for a certain amount of generations.”

Answer: Done (lines 109-110).

9. Line 82: “Replace “to an” with “in an” e.g. “... this process results in an increased ...”

Answer: Done (line 115)

10. Line 83: “Remove “the”. Replace “on” with “in” and “to a” with “in” e.g. “... fitness under artificial rearing conditions and in the same time in reduced performance under different rearing conditions ...” I think this sentence, in general, should be double checked.”

Answer: The whole sentence has been restructured and the meaning is now clearer (lines 115-116).

11. Line 86: “Include “as” “... factors such as temperature ...”?”

Answer: Done (line 119).

12. Line 87: “Again, the use of “biotypes”. See above (e.g. different medfly lines, populations, etc...)”

Answer: The word “biotypes” has been replaced with “populations” (line 119-120).

13. Line 91: “Replace “a-proteobacteria” with “α-proteobacteria””

Answer: Done (line 69).

14. Line 103-109: “In this part of the introduction I think it is important to mention how the kind of fitness alterations caused by Wolbachia onto its hosts can be dependent on how long the host is infected with Wolbachia. For example, new or recent infections more commonly result in higher fitness costs than long-standing ones. This will emphasize some differences between natural infections and transinfected lines, where Wolbachia was transferred from a donor to recipient recently. Otherwise, I agree that the relationship between Wolbachia and host represents a dynamic nature not simply explained by any one factor alone and warrants investigation of interactions with abiotic factors.”

Answer: We greatly appreciate the point raised by the Reviewer and we incorporated this suggestion to the respective paragraph (lines 85-91).

15. Line 125: “Citation “58” is wrongly placed here. Riegler & Stauffer, 2002 did not do laboratory crossings to establish level of CI. They analysed natural populations for infection frequencies. Plus, it is not clear if the authors refer to R. cerasi or C. capitata here.”

Answer: The reference is now corrected (line 155). 

16. Line 147: “Although the term “wildish” is explained in the Materials and methods section, I think it would be good to have a few additional words that explain what is meant by it already in the abstract or introduction. I may be wrong, but the term sounds a little awkward without further description.”

Answer: The term “wildish” is now also explained in Introduction (lines 109-110). See also comment No 8.

17. Line 148-152: “Sentence should be re-stated. Was it addressed or wasn’t it addressed? e.g. “By including Wolbachia-infected lines in our experiments, we addressed whether infection affects ...”

Answer: The sentence has been restructured and the aims of our study are now clearly stated clearly (line 181).

18. Line 158: “For reproducibility reasons, it may be worth noting what kind of lights were used (I’d assume this is more important for the adult rearing conditions).”

Answer: Specification regarding light quality are now given in lines 189-192.

19. Line 161: “Italicize the “w” in “wCer2””

Answer: Done (line 195).

20. Line 171: ““... for few months ...” May the authors provide a range or approximation of how many months in more exact terms?”

Answer: Actually both host fruit species were kept at 6 ± 2°C for 2 to 14 days before being used. This information is now corrected in the text (line 205).

21. Line 176: “Closing parentheses missing”

Answer: Corrected (line 210).

22. Line 194: “I think ImageJ may require a citation. Additionally, what was done with these measurements (length and width of pupae)? In the results, it appears that only survival and developmental times were recorded, so it is a little unclear to me what was done with these measurements and how they were incorporated into this study?”

Answer: We agree with the Reviewer’s comment. The pupae measurements were not used in the statistical analysis of the present paper. Hence the lines referring to the pupal dimensions are now deleted from the text (lines 232-235). 

23. “In the supplemental excel file, it is not clear to me how many samples and which samples belong to which of the 20 replicates (L195). This could be clearer. Maybe just by adding a total of how many individuals were finally used.”

Answer: The number of fruits (apples and bitter oranges) and of larvae implanted in fruits during our tests is now provided in the text (lines 237-240). The Supplementary excel file is provided as the raw data source containing all the information needed regarding pupation in each treatment.

24. Line 199-204: “Although I think the statistical procedures used in this study are sound, I would appreciate to read why they were chosen. Perhaps advantages, disadvantages, or even just references and what exactly was done using them (e.g. tests of significance, interactions, etc...). Also applies to the Wald test, which is only mentioned in the “Results” section (this could be included in the “Materials and methods” section, stating that it was used in the regression analyses to test for significance)”.

Answer: We thank the Reviewer for this comment and based on his suggestion we elaborated more on the reasons we chose the statistical methods used in this paper (lines 244-253) in the “Data analysis” section.

25. “I recommend adding sub-headers for each section within results and discussions for readability (maybe even materials and methods). E.g. survival rates, performance/developmental duration, etc... Otherwise, the results are presented clearly. The figures and tables are also well made and clear.”

Answer: Sub-headers were added to the “Results” and to the “Discussion” sections (see also Comment 2, Reviewer #1).

26. L217: Remove “the” “... expressed lower egg-to-pupa survival rates ...”

Answer: Done (line 269).

27. L344-348: “While I agree that the comparison of wildish flies to the non-infected Benakeio laboratory medfly line supports the idea that reduced genetic variation and an altered microbiome in laboratory lines may result in poorer survival and performance on host fruit, I think it is hard to rule out that the poorer performance of the 88.6 and S10.3 lines is due to the same reasons. Perhaps an additional bottleneck effect might have occurred while trans-infecting these lines and may be worth acknowledging that Wolbachia may not be the sole reason these lines under-perform, especially under higher temperatures where the influence of Wolbachia is supposed to be hampered.”

Answer: We thank the Reviewer for this comment. We also agree with the fact that both non-infected and Wolbachia-infected laboratory populations of medfly undergo major bottleneck pressures during domestication and artificial rearing. Hence Wolbachia infection may not solely account for the reduced fitness performance of these flies. We also added a phrase in the respective paragraph of discussion elaborating more this concept (lines 411-413). 

28. L438-441: “In accordance with the previous comment, if higher temperatures may be detrimental for Wolbachia infections and their influence on hosts, the fact that higher temperatures in this study affected the infected lines most may be indicative of microbiome deficiency or lack of genetic diversity, which the authors include. The authors mention how the microbiome hypothesis could be tested (L421). A genetic/genomic comparison to test the genetic variability hypothesis may be mentioned as well.”

Answer: We thank the Reviewer for this comment and we incorporated the suggestion made in lines 497-498 and 501-502.

29. L472-476: “These two sentences are not very clear. I’ve missed the take home message of Krainacker et al. Also, next sentence, I think it needs to be re-written for clarity.”

Answer: The reference to Krainacker et al. seemed rather out-of-focus and has been deleted (lines 543-545). The next two sentences have been rephrased to make content clearer. 

30. “I recommend a thorough inspection of the references section as some inconsistencies and mistakes seem to be present. Here are just a few examples: L611: Reference is missing journal name, volume and page numbers. L738: Reference is missing page number. L752: Journal name format is different from others. L766: Journal name format is different from others. L775: Journal name format is different from others. L786: Journal name format is different from others.”

Answer: The reference list has been thoroughly inspected and all inconsistencies have been corrected.

Thank you for considering this paper for publication.

Sincerely yours,

Nikos Papadopoulos, Professor

---

## [Decision Letter · Decision Letter 1]

13 Feb 2020

Effect of host fruit, temperature and Wolbachia infection on survival and development of Ceratitis capitata immature stages

PONE-D-19-25021R1

Dear Dr. Papadopoulos,

We are pleased to inform you that your manuscript has been judged scientifically suitable for publication and will be formally accepted for publication once it complies with all outstanding technical requirements.

With kind regards,

Luciano Andrade Moreira, PhD

Academic Editor

PLOS ONE

Additional Editor Comments (optional):

Reviewers' comments:

Reviewer's Responses to Questions

**Comments to the Author**

1. If the authors have adequately addressed your comments raised in a previous round of review and you feel that this manuscript is now acceptable for publication, you may indicate that here to bypass the “Comments to the Author” section, enter your conflict of interest statement in the “Confidential to Editor” section, and submit your "Accept" recommendation.

Reviewer #2: All comments have been addressed

2. Is the manuscript technically sound, and do the data support the conclusions?

Reviewer #2: Yes

3. Has the statistical analysis been performed appropriately and rigorously? 

Reviewer #2: Yes

4. Have the authors made all data underlying the findings in their manuscript fully available?

Reviewer #2: Yes

5. Is the manuscript presented in an intelligible fashion and written in standard English?

Reviewer #2: Yes

6. Review Comments to the Author

Reviewer #2: (No Response)

7. PLOS authors have the option to publish the peer review history of their article (what does this mean?). If published, this will include your full peer review and any attached files.

Reviewer #2: Yes: Vid Bakovic

---

## [Editor Report · Acceptance letter]

4 Mar 2020

PONE-D-19-25021R1 

Effect of host fruit, temperature and *Wolbachia* infection on survival and development of *Ceratitis capitata* immature stages 

Dear Dr. Papadopoulos:

I am pleased to inform you that your manuscript has been deemed suitable for publication in PLOS ONE. Congratulations! Your manuscript is now with our production department. 

With kind regards,

on behalf of

Dr. Luciano Andrade Moreira 

Academic Editor

PLOS ONE